# Club-like receptors respond to light touch but not to whisking

Taiga Muramoto [1], Takahiro Furuta [2] ✉, Taro Koike [3], Knarik Bagdasarian[4], Sotatsu Tonomura [5], Aya Takenaka[2], Yosky Kataoka[6,7], Mitsuyo Maeda[7,8], Asami Eguchi[6,7], Masaaki Kitada [3], Kenzo Kumamoto[1], Ehud Ahissar [4] ✉ & Satomi Ebara [1,2] ✉

Rodents explore their environment by actively whisking and contacting objects with their whiskers. Each whisker follicle contains hundreds of mechanoreceptors, but how specific afferents distinguish between self-motion and touch remains unclear. Here, using artificial whisking in male rats, intra-axonal recordings, and morphological reconstruction, we identify a distinct mechanoreceptor subtype—club-like endings—that respond exclusively to touch and remain silent during whisking. In contrast, Merkel and lanceolate endings exhibit mixed selectivity. Club-like endings are arranged in a single-layer circular array near the center of mass of the whisker-follicle unit, embedded in a collagen-rich structure called the ringwulst. Using scanning electron microscopy, we show that these endings are tightly anchored to the glassy membrane and collagen fibers, forming a mechanically isolated zone. This configuration minimizes activation during whisking while preserving sensitivity to touch. We propose that these features evolved to enhance tactile precision in whisking species, as supported by the absence of such specializations in non-whisking animals such as cats.

Natural perception is primarily active—animals control their sensory organs as part of perceiving their environments. While offering significant advantages over passive sensing[1], active sensing has computational costs. One of these costs is the need to differentiate changes occurring due to self-motion from those resulting from the interactions of the moving sensors with the environment[2]. One of the active-sensing modes that has been extensively studied is that of active vibrissal touch[3,4]. Recent work has emphasized the dual modes of vibrissal information acquisition: generative, where the animal sweeps its whiskers forward and backwards, and receptive, where the animal keeps its whiskers still, and collects mechanical signals from the relative motion between the head and an object[5].

Generative vibrissal sensing introduces a major mechanical challenge—allowing selective sensitivity to light touch. As the whiskers constantly move, such selectivity necessitates differentiating mechanical deformation induced by whisker-object touch from that induced by vigorous whisking in air. Since mechanoreceptors (MRs) have evolved to remarkable sensitivity (down to deformations of 10 nm)[6], the major challenge lies in preventing MR activation during whisking in air. Nevertheless, evolution has risen to this challenge. Using artificial whisking (achieved by electrical stimulation of the facial nerve in anesthetized rats), it was found that individual primary afferents from these follicles respond selectively either to whisking (W) or touch (T), or non-selectively to both (WT)[7,8]. Thus, W cells are

[1]Department of Anatomy, Meiji University of Integrative Medicine, Nantan, Kyoto, Japan. [2]Department of Systematic Anatomy and Neurobiology, Graduate School of Dentistry, The University of Osaka, Suita, Osaka, Japan. [3]Department of Anatomy, Kansai Medical University, Hirakata, Osaka, Japan. [4]Department of Brain Sciences, Weizmann Institute of Science, Rehovot, Israel. [5]Department of Anatomy, Kawasaki Medical School, Kurashiki, Okayama, Japan. [6]Graduate school of Science, Technology and Innovation, Kobe University, Kobe, Hyogo, Japan. [7]Laboratory for Chemical Biology, RIKEN Center for Biosystems Dynamics Research (BDR), Kobe, Hyogo, Japan. [8]Japan Electron Optics Laboratory (JEOL) Ltd., Akishima, Tokyo, Japan. ✉e-mail: furuta.takahiro.dent@osaka-u.ac.jp; ehud.ahissar@weizmann.ac.il; s_ebara@meiji-u.ac.jp

sensitive to whisking and insensitive to light touch (i.e., touch that does not block the movement of the follicle); T cells are insensitive to whisking in air and respond upon contact or detachment and WT cells are sensitive to both.

During generative sensing, the mechanical activation of the whisker follicle is shaped by a combination of kinematic and force-related variables—such as whisker angle, velocity, and bending moment—making the selective coding of specific stimulus features challenging[9–11]. Each whisker follicle contains hundreds of MRs, typically categorized into five types: Merkel, lanceolate, club-like, Ruffini-like, and free nerve endings[12–15]. Recently, genetic marking has enabled the recording from primary afferents of specific MR types[16,17]. Recordings from identified Merkel afferents during self-generated whisking in awake mice showed that Merkel receptors encode both whisking and touch, with some afferents responding predominantly to touch, some to whisking motion, and others to both[17]. In parallel, intracellular or intra-axonal recording and labeling from MR neurons have allowed the associations of spatial selectivities of responses of single MR neurons with their precise receptor morphology and location within the follicle[18,19]. However, these methods have not yet been applied to study receptor selectivities during active sensing.

In this study, we combined artificial whisking with intra-axonal recording and labeling methods to identify the type and location of individual MRs along with their functional selectivity. We found that club-like endings respond selectively to touch. Furthermore, we explored the morphological adaptations that enable this remarkable selectivity and found that the rat's follicle possesses a unique mechanical design that maximizes sensitivity to whisker touch while minimizing sensitivity to whisking in air.

## Results
### Selectivity of the mechanoreceptors during artificial whisking
We recorded 201 primary afferents from 104 anesthetized rats (males, ages 10 to 15 weeks old, weighting 250 to 350 g, SLC, Hamamatsu, Japan) during artificial whisking in air and against objects. Artificial whisking was applied at 5 Hz during trains of 2 s (10 whisks), with inter-train intervals of 2 s (Fig. 1b; see Methods). Intra-axonal recordings were done using glass electrodes filled with biotinylated dextran amine (5% BDA) or neurobiotin (20% NB) for labeling (see Methods). The neuronal tracers were injected intra-axonally to each of the recorded axons (Fig. 1a).

Stable recordings were difficult to obtain during artificial whisking and varied in duration. Of the 201 afferents, only 18 met the criteria of being recorded for at least one train of artificial whisking in air (active whisking), three trains against objects (active touch), and having fully reconstructed morphology (Fig. 2a, Supplementary Fig. 1). In total, one rete ridge collar (RRC)-Merkel ending, five club-like endings, six ring sinus (RS)-Merkel endings, and six lanceolate endings met these criteria and are reported here (Table 1).

In addition to identifying their type and location within the follicle (Fig. 2a), each primary afferent was classified as either a W, T or WT cell (Table 1) according to the difference between its response to whisking against an object and to whisking in air. W cells responded equally to both (Fig. 2b, cell No.14), T cells responded only upon touch (Fig. 2b, cell No.3 and 8) and WT cells responded differently to both (Fig. 2b, cell No.1 and 10). Note that (Fig. 2b) cell No.3 responded to the onset of contact, "Contact T cell"[7], whereas cell No.8 responded to detachment, "Detach T cell"[7]. Herein we do not distinguish between different subtypes of T cells. Quantitatively, their Touch Index (TI = (St-Sw)/(St+Sw), where St = spike count upon touch and Sw = spike count during whisking in air, see Methods) was previously found to be between [−0.2, 0.2] for W, larger than 0.8 for T, and otherwise for WT cells[20]. With our current intra-axonal recordings, almost all T cells showed TI = 1 (Table 1).

The three cell types recorded here, Merkel, lanceolate, and club-like endings, differed in both the distributions of their locations within the follicle (Fig. 2a) and the distributions of their response types (Figs. 2b, 3). Whereas Merkel and lanceolate endings were sensitive to both whisking and touch, club-like endings were sensitive only to touch (Table 1, Fig. 3).

### Morphological adaptations of the club-like endings
The stereotypic T response of the club-like endings suggests that they were selectively adapted through evolution to avoid responding to whisking in air. Given the remarkable sensitivity of mechanosensitive molecules to minute mechanical deformations (on the scale of 10 nm)[5], resisting responses to whisking in air requires precise adaptation. Therefore, we examined the morphological adaptations of the club-like endings.

The club-like endings are arranged as a one-layer ring around, and very close to the whisker shaft, covering all shaft angles except for the dorso-caudal 45 degrees[18,21]. Every whisker is innervated by ~60 club-like endings, each of them is attached to a particular shaft angle (Supplemntary Fig. 1b.). The one-layer ring of the club-like endings is located near the center of mass of the whisker-follicle unit, which is moved during whisking in air by the moment generated by a pair of intrinsic muscles, one attached to the follicle near its bottom and one near its top; the center of mass is very close to the floating pivot of whisker rotation, a point that moves the least during whisking in air[22,23]. Thus, the location of club-like endings is the best for avoiding responding to whisking in air.

Each axon terminal is covered by terminal Schwann cell (TS) sheaths (Fig. 4). Using semi-thin array tomography[24], we show that, consistently with previous studies[18,19], club-like endings possess short axon terminals embedded within a dense network of fine collagen fibers occupying core region of the C-shaped Rw (Figs. 5a–e, f, g, i, 6, Supplementary Movie 1). Thin processes of the TS sheath extended upwardly as a long longitudinal process (Fig. 5f, g, i–k). The TS sheath covering the club-like endings were sandwiched between dense thin fibers connecting with the glassy membrane and another set of thin fibers connecting with thick collagen bundles ascending from the marginal zone of the Rw (Fig. 5e–h). The bundles were collagen I-immunopositive and showed typical banding pattern (Fig. 5d, h, i). This tight TS-collagen connection was observed only with club-like endings (Fig. 4), adding another factor isolating mechanosensitive channels, such as piezo II[16,17], on these endings from movements that are common to the follicle and the whisker shaft encompassed by it.

The third factor found to potentially contribute to the decoupling of club-like endings from tissue deformations during whisking in air was the structure of the Rw in the rat and its attachment to the glassy membrane surrounding the epithelial follicle. The Rw is suspended from a curved edge of the glassy membrane (Fig. 6). The mass of the Rw determines the extent of inertial damping of the movements during whisking in air: the larger the mass the smaller the movement at the neck of the Rw, where the club-like endings are located, should be. This damping effect is analogous to that achieved using pendulum-type mass dampers in engineering applications, such as those used to suppress vibrations in tall buildings[25]. The operating principle involves suspending a mass (forming a pendulum) within the structure, allowing it to swing freely. When the structure undergoes lateral motion, the pendulum oscillates out of phase with the movement, thereby exerting an inertial counterforce that reduces the amplitude of the structural vibrations (see Supplementary Movie 2).

To test whether the mass and specific morphology of the Rw and its attachment to the epithelial follicle are unique to whisking animals, we compared the anatomy of the Rw in rats and cats (Fig. 6). The differences are striking: in the cat, (i) the Rw lacks a dense array of thick collagen fiber bundles, (ii) the glassy membrane is not curved, (iii) there is no lumen between the glassy membrane and the Rw body, (iv)

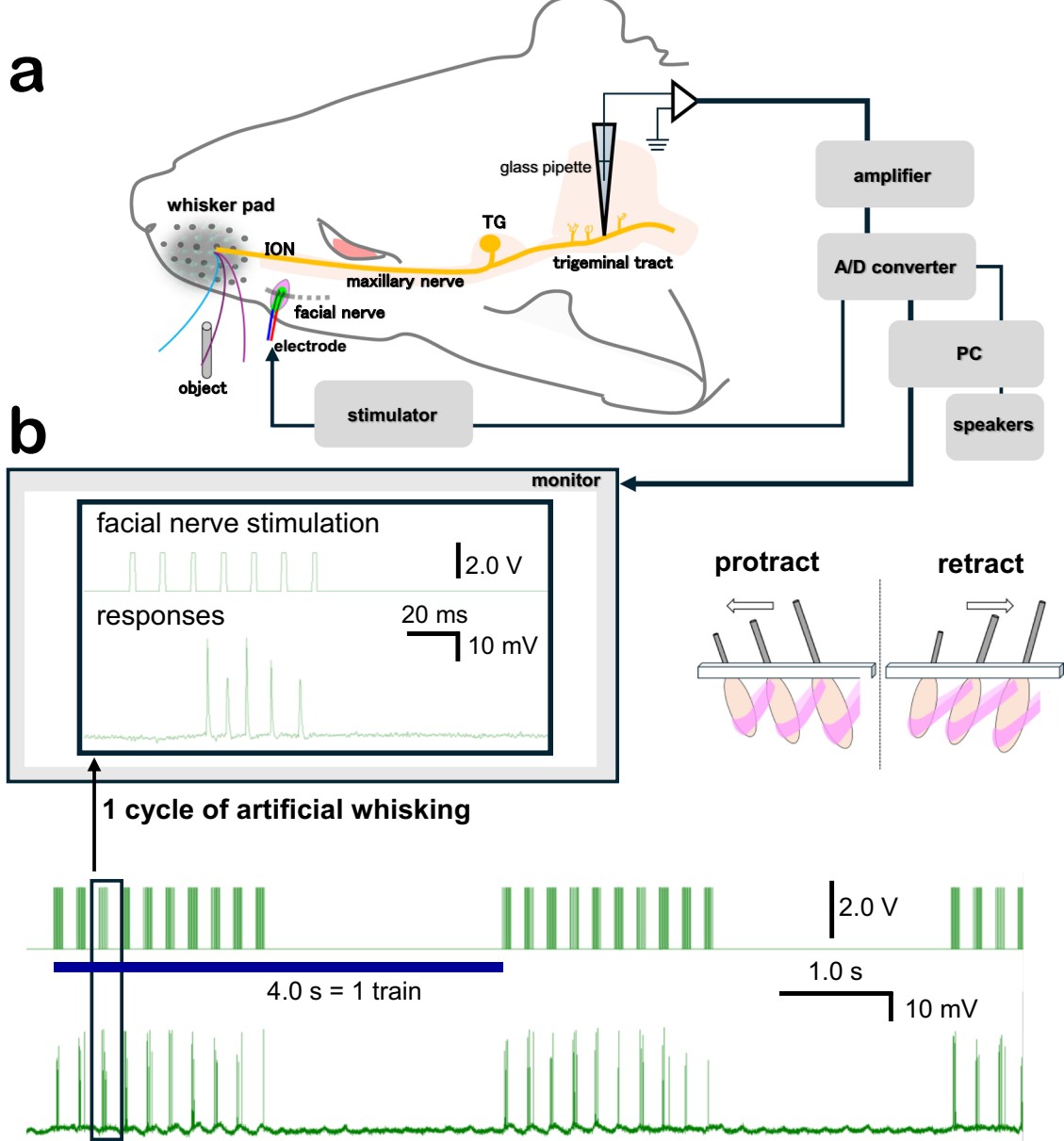

**Fig. 1 | Intra-axonal recording during artificial whisking. a** Intra-axonal recording and labeling. **b** Induction of artificial whisking at 5 Hz. Each whisking cycle lasted 200 ms, where protraction was actively induced by facial nerve stimulation (7 pulses at 83 Hz) and retraction was passive. Each stimulation train lasted 4.0 s and consisted of 2.0 s of artificial whisking (10 cycles at 5 Hz) and 2.0 s pause.

the neck of the Rw is also innervated by Merkel endings, which intermingle with club-like endings (Fig.6, Supplementary Fig. 2)[26] (v) the club-like endings are unevenly arranged at the middle part of the Rw, not forming a one-layer ring.

As a result, the Rw in cats provides less effective inertial resistance to whisking than in rats due to its smaller mass and different attachment to the glassy membrane. The absence of Merkel ending innervation in the rat is analogous to the absence of interfering structures—such as additional innervations and blood vessels—in the anatomical fovea of the primate retina. In both cases, this lack of interference suggests an adaptation optimized for high precision and sensitivity.

## Discussion

Club-like endings form a highly specialized class of receptors in the whisker follicle. They are arranged in a unique structure—a single-layer ring around the whisker shaft, protected in a unique manner by Schwann cells, embraced by a unique collagen structure—the Rw[27],

innervated in a 1:1 manner—one axon per one ending—and exhibit the smallest sizes of receptive fields among all follicle receptors[14,15,18,19,21,28]. Here we revealed their unique role in the encoding of active touch—during whisker protraction, club-like receptors respond selectively to active contacts and ignore active protraction in air. We revealed it using artificial whisking, in which protraction is active and retraction is passive[29].

Selectivity to active contacts during whisking requires a delicate mechanical selectivity. Piezo channels, responsible for detecting contacts in most mechanoreceptors, are sensitive to deformations as small as several nanometers[16,30]. Active whisking is a vigorous movement involving extrinsic and intrinsic muscles and rotating the follicles at high speeds and high accelerations[30]. We show here that evolutionary adaptation selected the best position for minimizing deformations during whisking in air—the whisker's center of mass—for localizing the club-like endings. This may explain the single-layer ring structure, keeping all ~60 endings near the center of mass. We further

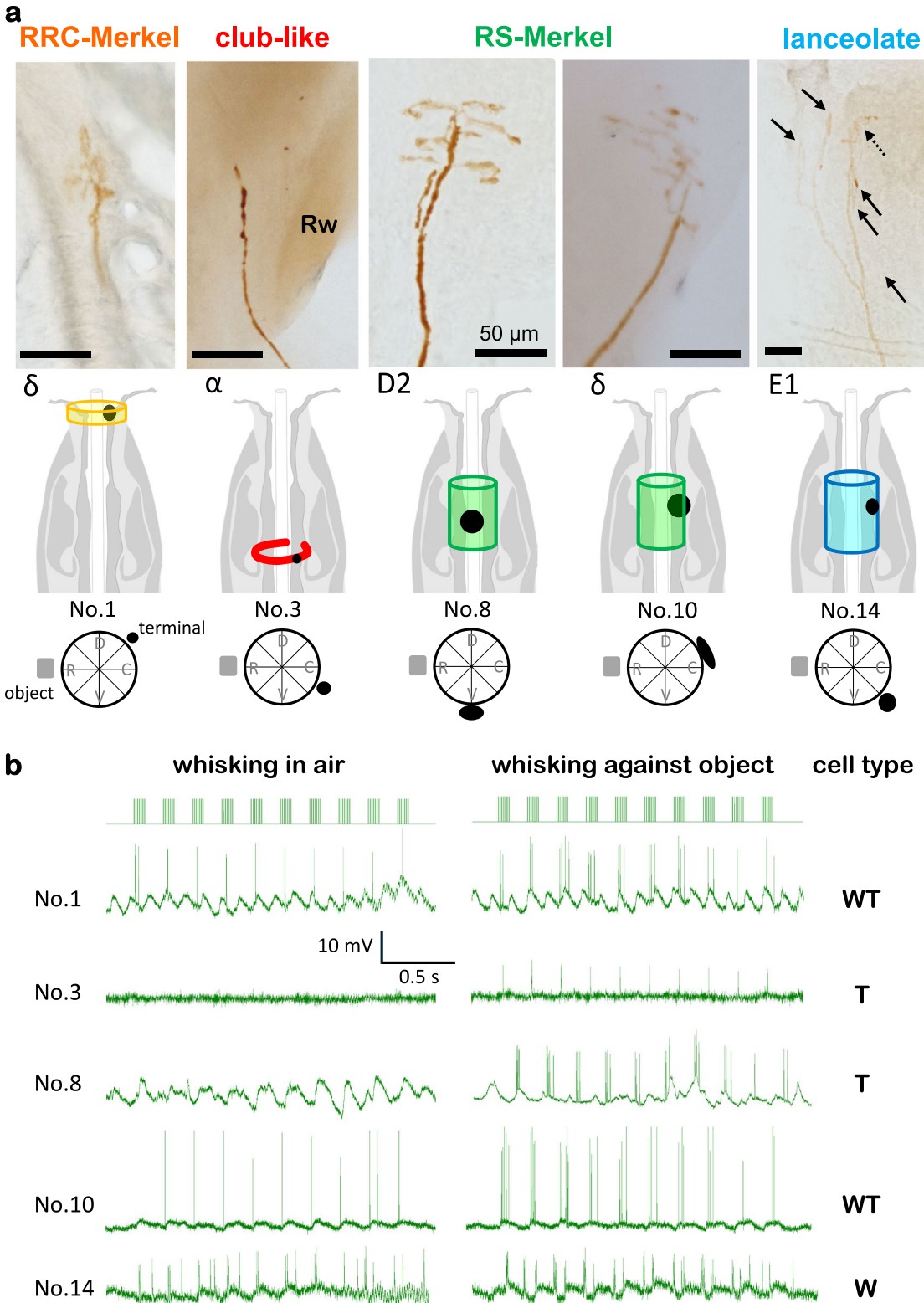

**Fig. 2 | Receptor identification and intra-axonal responses. a** Morphological visualization mechanoreceptors, 5 examples shown: RRC-Merkel (No.1), club-like (No.3), RS-Merkel (No.8 and 10), and lanceolate (No.14) (see Table 1). Scale bar, 50 μm. Black dots, location of the ending(s) along the longitudinal axis of the follicle. Color cylinders or ring, the area where other endings of the same type are distributed. Black dot or elongated shape, azimuthal location of the mechanoreceptor ending(s). The object (gray square) was placed rostrally to the whisker. **b** Responses of the neurons shown in a during 10 whisking cycles. Right, their inferred response types.

**Table 1 | Profiles of all neurons**

| ending | No. | a whisker | b Touch Index | c cell type | d recording time (s) | e Nspikes | f classification reliability |
|---|---|---|---|---|---|---|---|
| RRC-Merkel | 1 | δ | 0.54 | WT | 66 | 285 | 0.9 |
| club-like | 2 | E1 | 1.00 | T | 58 | 235 | 1 |
| | 3 | α | 1.00 | T | 115 | 125 | 1 |
| | 4 | D7 | 1.00 | T | 158 | 278 | 1 |
| | 5 | α | 1.00 | T | 205 | 453 | 1 |
| | 6 | C3 | 1.00 | T | 125 | 150 | 1 |
| RS-Merkel | 7 | E7 | 1.00 | T | 40 | 160 | 1 |
| | 8 | D2 | 1.00 | T | 32 | 158 | 1 |
| | 9 | C2 | −0.09 | W | 140 | 391 | 0.8 |
| | 10 | δ | 0.43 | WT | 300 | 920 | 0.8 |
| | 11 | C2 | 1.00 | T | 155 | 279 | 1 |
| | 12 | C2 | 1.00 | T | 155 | 470 | 1 |
| lanceolate | 13 | MV | 1.00 | T | 80 | 104 | 1 |
| | 14 | E1 | −0.13 | W | 58 | 313 | 1 |
| | 15 | C3 | 0.90 | T | 185 | 3032 | 0.9 |
| | 16 | B3 | 1.00 | T | 130 | 1037 | 1 |
| | 17 | β | 0.43 | WT | 280 | 1559 | 1 |
| | 18 | CH | −0.20 | W/WT | 190 | 88 | 0.5 |

RRC-Merkel: No. 1, Club-like: 2–6, RS-Merkel: 7–12, and Lanceolate: 13–18. **a** Whisker label. MV, micro vibrissa; CH, common hair; Latin letter, straddler; Xn, whisker in row X and column n. **b**, **c**. Cell types as identified by the Touch Index (TI). WT, Whisking/Touch cell; T, Touch cell; W, Whisking cell. (See Methods). **d** Recording time. **e** total number of recorded spikes. **f** classification reliability.

hypothesize that adding extra care of Schwann protection and inertial damping by the Rw likely complete the supreme isolation of club-like ending from whisking in air deformations.

The circular arrangement of club-like endings raises an intriguing hypothesis: these endings may not only detect contact but also encode its azimuth relative to the snout. This potential azimuth coding is facilitated by the circular organization of club-like endings, which form a single-layer ring around the whisker shaft of about 60, nearly identical, endings. Consequently, the first ending to respond upon contact is likely the one experiencing the strongest deformation.

Upon contact, the deformation induced by the bending moment is expected to be non-uniform around the shaft due to the uneven distribution of contact stress. This leads to a key coding principle: because whisker torsion (rotation around its own axis)[31,32] correlates with azimuthal protraction, the location of maximal deformation will vary with the azimuthal position of the contact. Thus, the identity of the first responding club-like ending encodes the azimuth of contact, with a resolution on the order of protraction amplitude divided by 60.

The whisker azimuth is also encoded via a continuous representation of the whisking phase by primary afferents[7,33]. In awake mice, whisking phase was shown to be encoded with high resolution by follicle-associated Merkel afferents, and with lower precision by cutaneous and supraorbital mechanoreceptors[17,34]. Furthermore, in awake behaving rodents, the firing of primary afferents was found to be best predicted by a combination of kinematic (e.g., angle, velocity) and dynamic (e.g., bending moment) variables, making the direct interpretation of their responses ambiguous[9,10]. Here, we propose that combining re-afferent (self-motion) signals from Merkel and other afferents with ex-afferent (contact-driven) signals from club-like endings may help disambiguate azimuthal coding upon object contact. Whether rodents utilize either of these coding strategies, or a combination thereof, remains to be tested.

The precise arrangement of club-like endings may be instrumental in understanding their mechanical activation during contact. The torque developed along the whisker shaft during contact or during whisking against air resistance is largely similar[35]. Is the selectivity of club-like endings to contact determined solely by their exact location? Or is it also influenced by a sensitivity to axial whisker deflections, lateral non-bending loads, high-frequency vibrations, or stick-slip events[35]? These questions remain to be clarified in future experiments, aided by precise tracking of whisker kinematics.

The precise 1:1 innervation pattern and the central localization of the club-like endings along the follicle resemble the specialized adaptation of the primate retina. The distinct sensitivities of peripheral and foveal cells in the primate retina optimize their ability to detect scene-related and object-related features, respectively. Peripheral vision is crucial for efficient search, guiding eye movements toward areas of interest. In turn, foveal vision provides high-resolution analysis of the area of interest[36]. Similarly, the unique characteristics of receptors distributed along the whisker follicle may be adapted to optimize sensing during whisking in air by non-club-like endings located far from the center of mass, and sensing precise contact timings by club-like endings that are concentrated near the center of mass.

High-resolution selective sensitivity to touch necessitates evolutionary specializations that eliminate, or strongly dampen, mechanical deformations that can be induced during whisking in air. By comparing follicle morphology between non-whisking (cat) and whisking (rat) animals we identified five major differences that might represent such evolutional adaptation. Three of the differences increase the inertial resistance of the Rw, which contains the club-like endings ring in its neck, in the rat compared to the cat: increased mass, curving the glassy membrane, and creating a lumen between the Rw body and the glassy membrane. The fourth difference, indicating the cleaning the Rw's neck area from interfering structures in the rat, might be needed for coping with the ultimate challenge of high sensitivity together with high accuracy, similar to the cleaning of fovea centralis in the primate retina from interfering structures. The fifth difference, the formation of a one-layer ring in the rat, suggests an accurate metric coding by club-like endings.

The specialization of club-like endings was substantially different from those of the other receptor types studied here: Merkel and lanceolate endings. Individual Merkel and lanceolate endings are typically

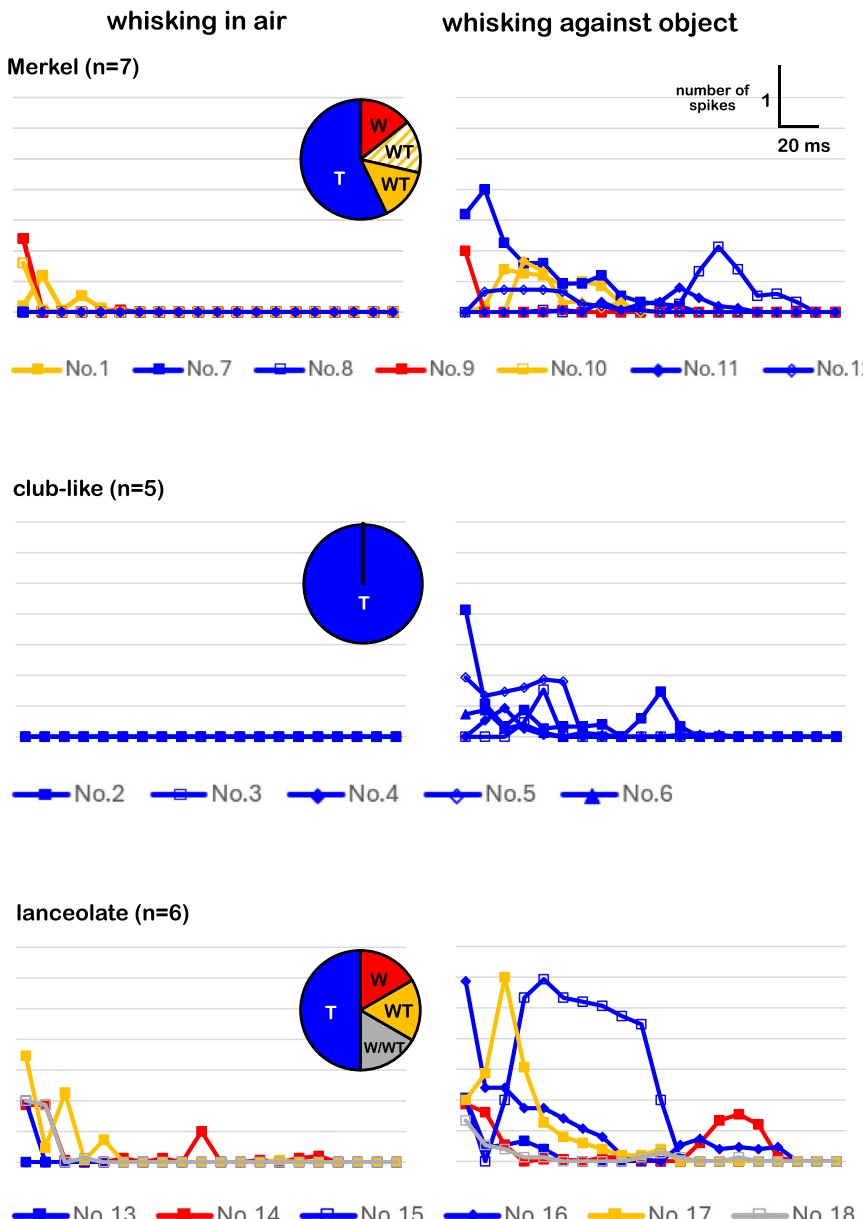

**Fig. 3 | Response selectivities.** All neurons were identified morphologically. Line graphs show the single whisk PSTHs of all recorded neurons ($n$ = 18). Blue, Touch (T); yellow, Whisking/Touch (WT); red, Whisking (W). Cell No 1. innervated RRC-Merkel endings and was a WT cell (shading on the pie chart). Scale bars appear at the top-right corner. Source data are provided as a Source Data file.

2 and 3-fold larger than club-like endings, respectively[15,21]. Their distribution in the follicle is broad and their surrounding tissues do not exhibit any highly specific morphological adaptations.

Merkel endings are located within the basal layer of the epithelial sheath, an area free of connective tissue. This positioning enables them to detect deformations in their vicinity, whether these are caused by whisking or touch. Their large variability in receptor size, number of endings per cell, location relative to the whisker shaft, and morphological relationships with their surrounding tissue[14,19], allows substantial variability in response selectivity, as observed here. Lanceolate endings are like lancets suspended in the loose space between the mesenchyme and the glassy membrane, called the intermediary zone, supported by Schwann cell processes[14,19,28]. Their vertical hanging morphology suggests different sensitivities than those of Merkel endings. Yet, nothing in their surrounding tissue, or location, prevents them from responding to mechanical deformations induced by either whisking or object contacts.

These comparisons suggest that while individual Merkel and lanceolate endings may exhibit selective sensitivities to whisking and touch, they were not selected as a group to address a specific sensory selectivity. In contrast, it seems that the special morphology of club-like endings and their surrounding tissue, including their one-layer ring arrangement around the center of mass of the whisker-follicle unit, was selected through evolutionary and developmental processes to address a specific need. This need is likely the detection of contact with external objects at a high confidence, i.e., at a high signal-to-noise ratio, and at a high temporal reliability. High temporal reliability is required for accurate object localization[3,31], thus an evolutionary postdiction of this study is that the special arrangement of club-like endings in whisking rodents evolved along with their ability to localize external objects at high accuracy.

This study shows, for the first time, morphological specialization that enables a specific functional selectivity in active touch. The challenge that this specialization solves is that of elimination of responses

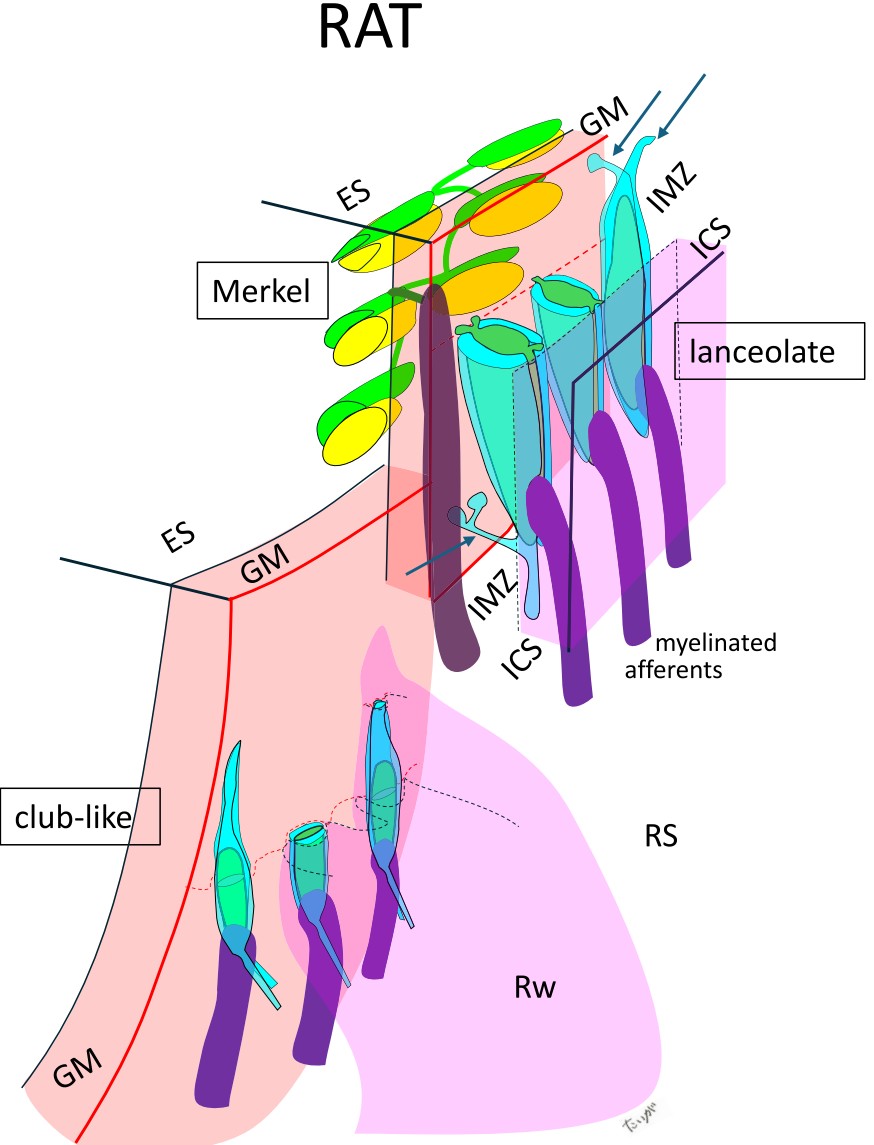

**Fig. 4 | Comparative morphological diagram of mechanoreceptors in the rat.** Individual club-like endings are composed of a small flat axon terminal (green) sandwiched between the two terminal Schwann cell sheaths. The processes extend upward. Each club-like ending is enclosed completely by the collagen mass of the ringwulst (Rw), which in turn "floats" in the ring sinus (RS). Lanceolate endings are consisting thickened lancet-like axon terminals holed between sides by the terminal Schwann cell sheaths. They are distributed within the intermediary zone (IMZ), where collagen fibers are scarce. Several terminal Schwann cell processes (arrows) are bridging between the glassy membrane (GM) and the inner connective tissue sheath (ICS) supporting the endings. Merkel terminal disks are embedded in the basal layer of the epithelial sheath (ES).

to whisking in air while allowing high sensitivity to contacts made during such whisking. The solution is localization of tiny receptor endings near the center of mass of the whisker-follicle unit, arranging them in a one-layer ring close to and around the shaft, protecting the receptor endings by a thin Schwann layer that is completely embedded within a dense network of thin collagen fibers and generating an inertial resistance by a specially-adapted collagen weight (Fig. 7). Whether similar adaptations accompany the refinement of active sensing in other modalities remained to be discovered.

## Methods
### Animals
Adult male Wistar rats (10 to 15 weeks old, weighting 250 to 350 g, SLC, Hamamatsu, Japan) were used in this study. Cat samples obtained in previous studies[14,26], using the same fixation methods as those described herein, were reprocessed and analyzed. The study was carried out in accordance with the Meiji university of Integrative Medicine

Animal Care and Use Committee (#2016-004, 2019-002, 2022-001, 2023-006,009).

### Intra-axonal recording and labeling
Under deep anesthesia (Isoflurane 1.5–2.0%, 0.5 L/min), rats were fixed in a stereotaxic apparatus while breathing freely. Body temperature was maintained at 37–38 °C by a heating pad. Intra-axonal recording and labeling of primary afferents were performed using the methods described in previous studies[18,19], here alongside facial nerve stimulation[7,8]. In brief, a small opening on the skull was made to expose the cerebellar surface, then a glass pipet filled with neuronal tracer was inserted to explore a proper axon in the trigeminal tract. The skull opening was located around 3.5 mm posterior and 3.2 mm laterally from Lambda, and the electrode was lowered 5–7 mm from the cerebellar surface. Polished glass pipette (Kwik-fill, World precision Instruments, Florida, USA) filled with biotinylated dextran amine (5% BDA MW:3000; Invitrogen, Eugene, OR), or neurobiotin (20% NB

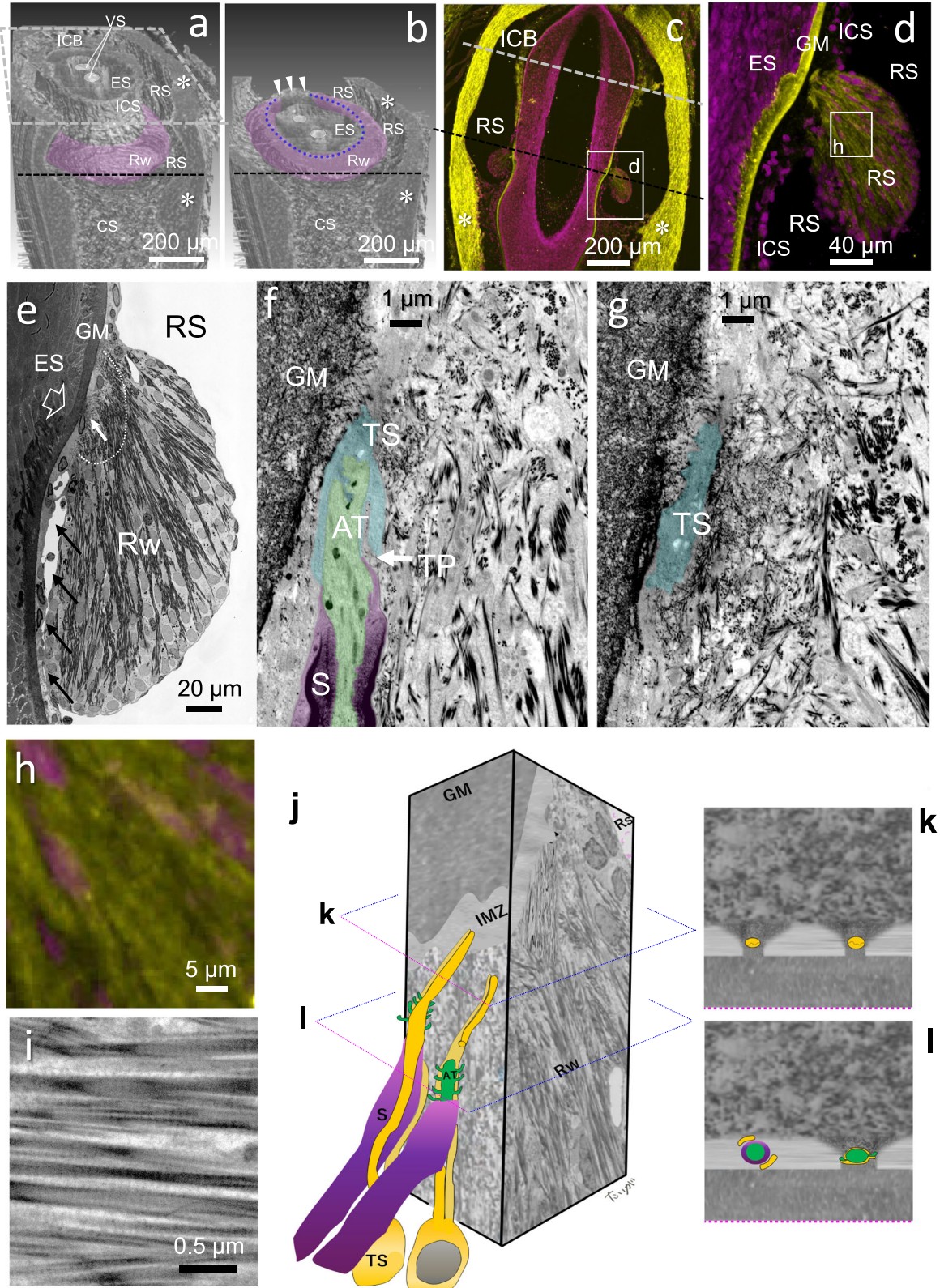

Vector Lab. Inc., Burlingame, CA USA) dissolved in 1 M potassium acetate, were used (electrode resistance: 50–150 MΩ, tip diameter: 1–2 μm). Resting potential immediately decreased when the pipette was successfully entered to a single axon. Then the receptive field of the vibrissa was characterized using a bamboo stick. After that, artificial whisking was applied. Recording signals were amplified (IR-183,

Cygnus Technology, USA) and A/D converter (Power Lab 8/30, AD Instruments, New Zealand, sampling rate: 2500 per second,) and an audio speaker, digital display, and computer with recording software (Chart 5, AD instruments) were used to monitor the responses. Recorded signals included low-frequency components (<20 Hz), likely reflecting physiological activity such as cardiac and respiratory

**Fig. 5 | Embedding of club-like endings within surrounding tissue. a, b, e–j** Scanning electron microscopy (SEM) of serial semi-thin sections. **a** The top surface, outlined by a dashed gray square, represents a cut plane at the level of the inner conical body (ICB), a connective tissue structure forming the ceiling of the ring sinus (RS) (see corresponding dashed gray line in **c**). *, indicates the thick collagenous capsule. CS cavernous sinus, VS vibrissal shaft. **b** The follicle is sectioned along a plane indicated by the black dashed line in **a** (see also **c**). The C-shape of the ringwulst (Rw) is clearly visible. The C-shape opening is located dorsocaudally (arrowheads). Club-like endings are distributed along the medial edge of the Rw (highlighted by a blue dotted line). **c, d** Immunohistochemical staining of collagen I (yellow) on a longitudinal frozen section. The Rw is filled with collagen I. Nuclei are counterstained in magenta with Pyridinium iodide. **e** A club-like ending (white

arrow) abuts the terminal curve (open arrow) on the basal face of the GM. The ending is embedded within a dense network of fine collagen bundles (enclosed by the dashed line). Thick collagen bundles radiate from the core of the Rw outward, toward the RS. **f, g** After removal of the myelin sheath (S) at the terminal point (TP), the axon terminal (AT) of the club-like ending is enveloped by a terminal Schwann cell (TS). The TS process extends upward and is embedded in fine collagen fibers. **h** Higher magnification of immuno-stained collagen I (yellow) with nuclei (magenta) of a part of Rw (d). **i** Electron microscopy of the collagen fibers of the thick bundles in the Rw showed striations. **j** Schematic diagram of the three-dimensional organization of two club-like endings (see Supplementary Movie 01). **k, l** 2D visualization of two cross sections marked in (**h**).

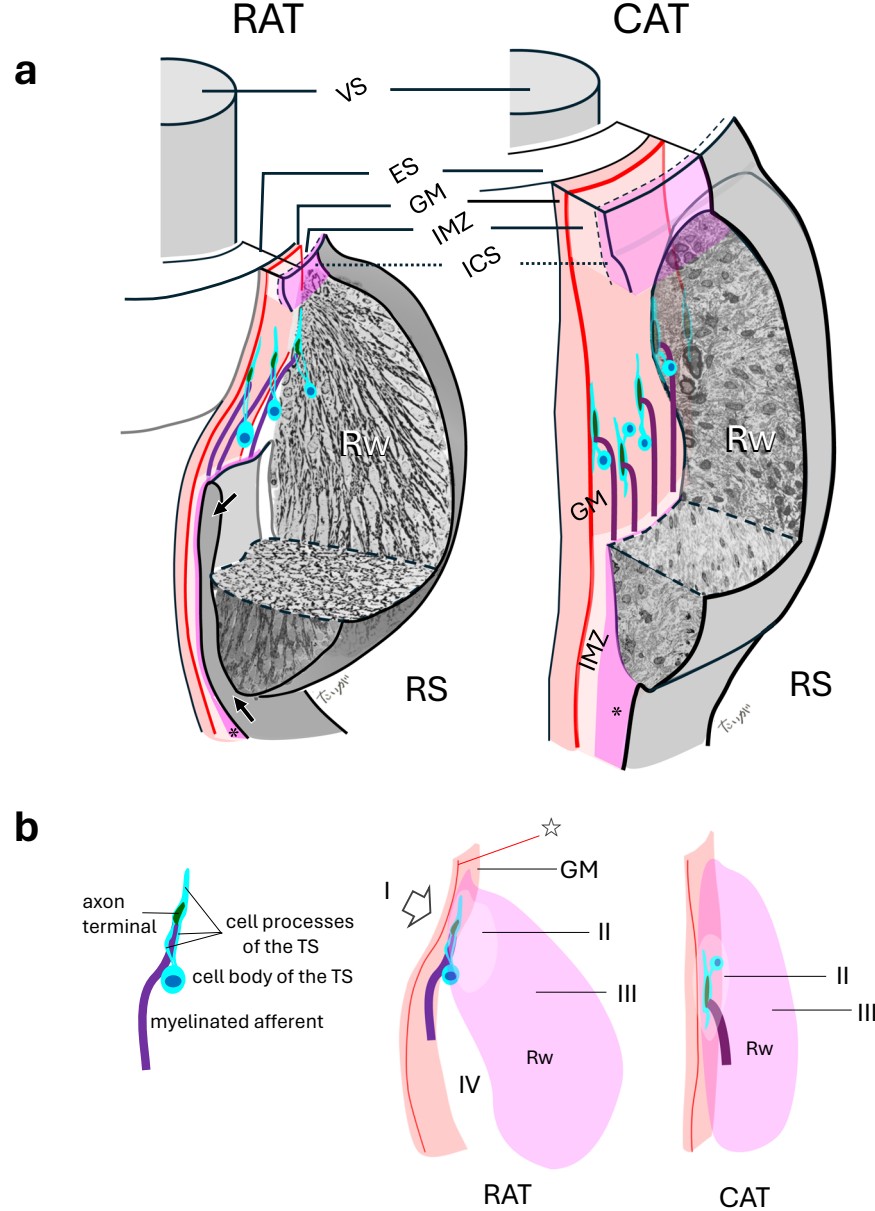

**Fig. 6 | Comparative organization of club-like endings and the ringwulst (Rw) in rats and cats. a** In rats, the open ring sinus extends deep into the Rw (arrows), whereas in cats it does not. Club-like endings in rats are distributed along the terminal curve (open arrow) of the GM. In contrast, cats lack a terminal curve, and their club-like endings are irregularly positioned within the mid-region of the Rw. Thick collagen bundles are regularly arranged in rats, while finer, less organized

fibers are observed in cats. VS, vibrissal shaft. ICS continues to *. **b** Simplified comparative schematics. A star indicates the basal border of the GM. I, terminal curve (open arrow), present only in rats; II, core region of the Rw containing embedded endings; III, outer region of the Rw; IV, lumen allowing continuity between the Rw and the ring sinus in rats.

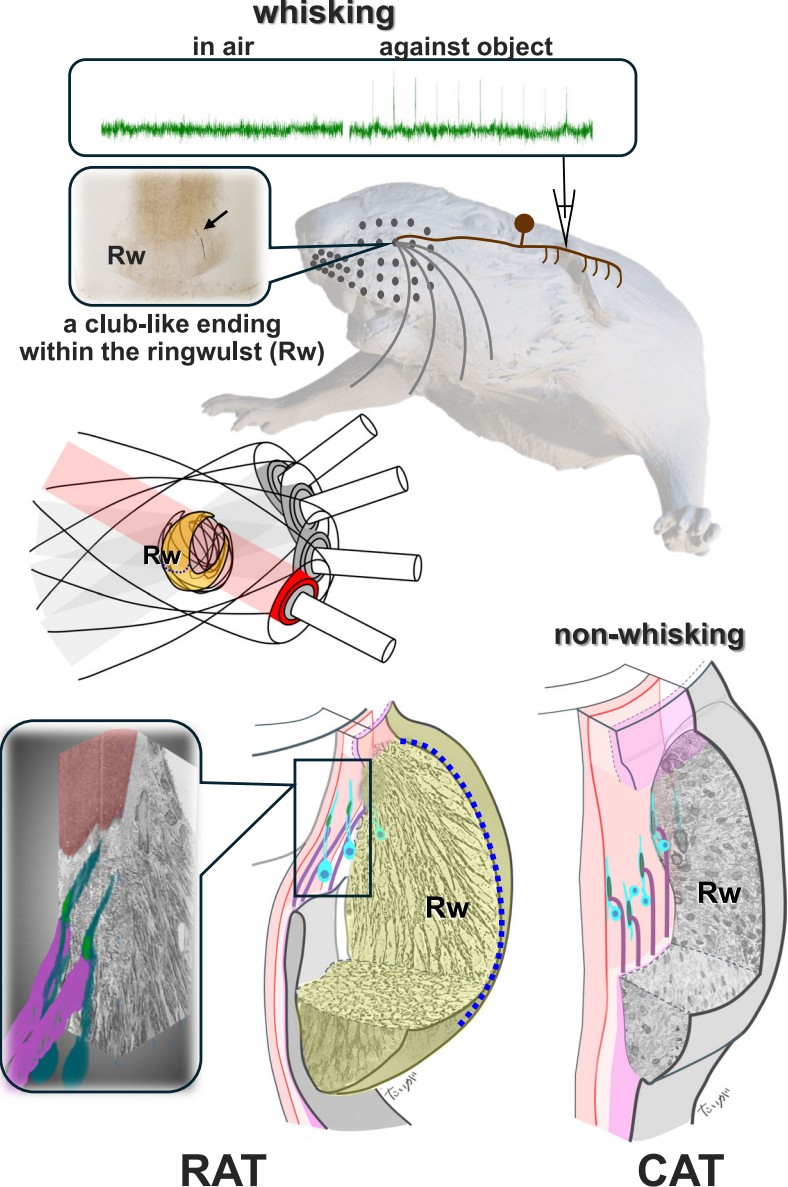

**Fig. 7 | Summary Figure.** How do animals sense external touch while their sensory organs are in constant motion? This study reveals an evolutionary adaptation in rat whisker follicles: specialized receptors structurally tuned to ignore self-motion and respond only to touch. The endings are club-like endings embedded within the ringwulst, a sausage-like shaped and collagen-rich structure, located near the center of mass of the whisker-follicle unit.

rhythms, and high-frequency components, primarily attributed to neural electrical activity. Recorded data were analyzed using dedicated software (Spike 2, Cambridge Electronic Design).

After completing electrophysiological recordings, neuronal tracer was electrophoretically injected into the single axon via the glass pipette (25–50 nA, 2 Hz, 15–20 min). Finally, the skin was closed by silk thread.

### Artificial whisking
Artificial whisking was induced by stimulating the facial nerves, as described in detail previously[37]. Briefly, bipolar, rectangular electrical pulses (7 pulses, 2.0 V, 2 ms duration) were applied to the distal part of the facial nerve through an isolated pulse stimulator (PG4000A, CYGNUS, Dorset, UK, BSI-950, Dagan MN, USA) at 83 Hz. Trains of 4 s were employed, each composed of 2 s of facial nerve stimulation (at 5 Hz) followed by a quiescent interval of 2 s. Artificial whisking were performed in air and against an object. The object (vertical pole of

3 mm diameter) was placed rostrally to the whisker, distanced from the skin by the equivalent of 70–90% of the whisker's length.

### Analysis of physiological data
Physiological data were analyzed using raster plots, post-stimulus time histograms (PSTHs) and computations of the Touch Index. Following Yu et al.[20], we quantified the responses during their steady states, which were typically stable during the last 6 cycles in each train.

Touch Index (TI) was calculated as $TI = (St-Sw)/(St+Sw)$, where $St$ = spike count upon touch and $Sw$ = spike count during whisking in air. $St$ and $Sw$ were quantified by calculating the firing rate during protraction, i.e., during the first 100 ms in each cycle. $TI = 0$ means that the neurons responded the same during whisking in air and whisking against object. $TI = 1$ means that the neurons only responded during whisking against object. $TI = -1$ means that the neuron was completely inhibited by the object. Following Yu et al.[20], we classified Whisking cells as showing $-0.2 <= TI < 0.2$, Whisking/Touch cells as showing

$0.2 <= TI < 0.8$ and Touch cells as showing $0.8 <= TI <= 1$; none of our cells showed $TI < -0.2$.

Classification reliability was evaluated by repeating the classification process 10 times, each time with only one train in air and one train against an object, randomly selected. Classification reliability (Table 1) was defined as the fraction of identical classifications among the 10 repetitions.

### Visualization of neuronal markers

After sufficient survival time, up to 20 h for neurobiotin or 10 days for BDA, the animals were deeply anesthetized (Isoflurane, 7% Chloral Hydrate) and then perfused transcardially with saline (0.9%NaCl, room temperature) followed by a fixative solution of formalin (10–20%) with 0.1 M sodium phosphate buffer (PB). Whisker pads were removed and then immersed in 30% sucrose in 0.1 M PB. Specimens were made into frozen serial sections (100–140 μm) by cryostat (Leica CM3050S). The maxillary nerve bundles were separated with forceps into tiny bundles.

All serial sections were rinsed in 0.1 M phosphate buffered saline containing 0.3% Triton-X100 (PBS-T) and then incubated in 3% hydrogen peroxide ($H_2O_2$) overnight. The tissues were rinsed several times in PBS-T, and then they were immersed in a horseradish peroxidase conjugated avidin-biotin complex (ABC, Elite, 1:300, Vector Labs, USA) in PBS-T at 4 °C for 1 day. After that the tissues were rinsed several times in PBS-T and reacted in a solution of diaminobenzidine (0.02%, DAB) including 0.01% cobalt-nickel ammonium sulfate added 0.3% $H_2O_2$ in 0.05 M tris-HCl buffer (pH 6.8). All serial sections were mounted on gelatin coated glass slides. Finally, tissues were enhanced by 0.05% osmium ($OsO_4$), dehydrated by ethanol and coverslipped with mounting medium (Entellan New, Merch-Aldrich Japan, Tokyo, Japan).

These samples were observed using a light microscope (i80, Nikon, Tokyo, Japan). High resolution digital photomicrographs were obtained with a DXM1200 camera (Nikon) using image analyzing software (NIS-Elements, BR, Nikon). (Supplementary Fig. 1)

### Immunohistochemistry

Whisker pads were horizontally processed into serial frozen thick sections as mentioned above. The follicles were processed immunohistochemically using goat collagen type I IgG antibodies (1:200, Arigo, Taiwan, ROC), Alexa Fluor 488 conjugated anti-goat IgG (1:300, Vector, USA) and pyridinium iodide for nuclei. The most perfectly stained samples were selected and submitted for observation by confocal scanning microscopy (Nikon-C1, Japan) (Fig. 5c, d).

### Scanning electron microscopic semi-thin array tomography

Rats were deeply anesthetized and perfused with 0.1 M PB followed by 4% formaldehyde (FA) and 0.05% glutaraldehyde (GA) in 0.1 M PB. Whisker follicle was removed and immersed in 4% FA in 0.1 M PB for 12 h at 4 °C. Whisker follicle was embedded in 3% agarose in 0.1 M PB and sliced with a linear slicer (45 μm-thick, Neo-Linear Slicer AT, Dosaka EM, Kyoto, Japan). Agarose was removed and the sections were fixed with 2% FA and 2% GA in 0.1 M PB for 10 min at room temperature followed by $OsO_4$ in 0.1 M PB at 4 °C, and then dehydrated with ascending concentrations of ethanol solution and embedded in epoxy resin. Serial semi-thin sections (250 nm thickness, Leica, Wetzlar Germany) were mounted on a piece of silicon wafer[24]. The sections were stained with 1% uranyl acetate for 15 min followed by Sato's lead staining solution for 5 min[38].

The sections were observed by field-emission SEM using JSM-IT800 (JEOL, Tokyo, Japan) or SU8600 (HITACHI, Tokyo, Japan), were aligned the XY axes and segmented manually using dedicated software (Photoshop 2024, Adobe System Co., Ltd., Tokyo Japan). 3D reconstructions were made using a dedicated software (Dragonfly, Comet Technology Canada Inc, Canada) (Fig. 5, Supplementary Fig. 2, Supplementary Movie 1).

### Reporting summary

Further information on research design is available in the Nature Portfolio Reporting Summary linked to this article.

### Data availability

All data including Supplementary files and Source Data files of this study are provided to the paper. Source data are provided with this paper.

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

## Acknowledgements

We thankfully appreciate to Dr. Sebastian Haidaliu, Dr. Eldad Assa, and Dr. Guy Nelinger (Weizmann Institute of Science, Israel), Dr. Mari Hirose, Dr. Tomokazu Murase (Meiji University of Integrative Medicine, Japan), Dr. Yumi Tsutsumi (Hyogo Medical University, Japan), Dr. Fumihiko Sato (The University of Osaka, Japan) for scholar and professional discussions and excellent technical support. We also thank Keiko Okamoto-Furuta (Kyoto University, Japan) for helpful discussion in electron microscopic analysis, Biomedical Central Research Center (Kansai Medical University, Japan) for the use of an ultramicrotome, Mr. Katsuhiko Taki (Nihon Visual Science Inc., Japan) for imaging technical support, and Dr. Edward L. White (Ben-Gurion University, Israel) for a continuous generous support. This work was supported by KAKENHI (JP21H03529, JP23K21711, and JP22K19403 to T.F., and JP23K06311 and JP22H04926 "Advanced Bioimaging Support (ABiS)" to S.E.) from the Japan Society for the Promotion of Science (JSPS). This study was also supported by the Japan Agency for Medical Research and Development (AMED) (JP23dm0207112 to T.F.) and Kobayashi Foundation (#384) to S.E.

## Author contributions

T.M. designed the study, conducted the experiments, analyzed the data, generated the figures, and wrote the paper. T.F. developed the intra-axonal recording and labeling and wrote the paper. A.T. pursued the intra-axonal recording and labeling and contributed to writing. S.T. and K.B. established the intra-cellular recording and labeling with artificial whisking and contributed to writing. T.K., Y.K., M.M., A.E., M.K., and K.K. pursued electron microscopic studies and contributed to writing. E.A. designed the study and wrote the paper. S.E. designed, led, and supervised the study and wrote the paper.

## Competing interests

The authors declare no competing interests.
