## [Transparent Peer Review file · Nature Communications]

Club-like Receptors Respond to Light Touch but not to Whisking

Corresponding Author: Professor Satomi Ebara

Version 0:

Reviewer comments:

Reviewer #1

(Remarks to the Author)

In their paper 'Club-like Receptors Respond to Light Touch but not to Whisking' Muranoto et al. investigate structure function relationships of vibrissa afferents. Specifically, the authors apply intra-axonal recordings combined with neurobiotin-labeling, electron-microscopy and electric whisking paradigms, an astounding range of analysis technologies.

The authors identify a range of afferents including significant numbers of vibrissal Merkel-receptors, lanceolate-endings and club-like endings. The highlight of the study is the author's conclusion that club-like endings are so-called touch cells, i.e. cells that do not respond to whisking alone but are responsive to object contact.

The authors also do beautiful electron-microscopy work and reveal novel details about the anchoring of club-like endings, which they show to be attached to the glassy membrane and also being anchored by collagen fibers. On grounds of these morphological reconstructions and by based on additional comparative evidence (comparisons to non-whisking cats) the authors argue that the ringwulst might actually contribute to dampening whisking related responses in club-like afferents. I think the whisking-response-dampening by the ringwulst is an interesting hypothesis, but the authors present it as a fact, which I think is putting this idea too strongly.

Major Points

1. This is an important paper. The authors are pioneers in understanding the vibrissa follicle and whisker response properties. The intra-axonal recordings performed here in whisking animals are nontrivial, but absolutely decisive for understanding the vibrissal system. Classifying club-like endings as touch-cells is an important advance that fully justifies publication.
2. What do the authors think is origin of the prominent subthreshold activity seen in many afferents (Figure 2)? I am asking, because the authors penetrate cells a centimeter or more away from the mechanosensory tip of the afferents and I expected that we should see little subthreshold signal as a result of axonal filtering.
3. I think the ms will benefit if the results followed a bit more tightly the Figures and individual panels. In general, the paper is well written, but the detailed link to the Figures would often be advantageous.
4. As I said earlier the whisking-response-dampening by the ringwulst is an interesting hypothesis. It would be great if the authors could explain thinking in more detail, perhaps adding a schematic as to how the ringwulst is supposed to do this trick. Also, I am not sure the idea is correct and the authors need to present this more cautiously.
5. I enjoyed the careful electron-microscopy.

Minor

1. Make sure all abbreviations are explained in all legends; I could not find IMZ.
2. Figure 5 C, D please show higher magnification of the immunofluorescence.

Reviewer #2

(Remarks to the Author)

This study is commendable for tackling a fundamental and unresolved question in sensory neuroscience: the structure-function relationships in low-threshold mechanoreceptors (LTMRs). The authors employ a technically demanding approach — intra-axonal recordings — to address this issue, and they present valuable data on whisker LTMR responses during artificial whisking, along with anatomical insights.

However, I have concerns regarding the sample size and the generalizability of the findings to natural whisking behavior.

Major Comments

Line 94: Were the recordings of sufficient duration to yield robust estimates of the response index for each afferent? Depending on the number of spikes in each recording, the classification of an afferent could potentially shift based on the presence or absence of few spikes. Please provide more specific information on the duration of each recording (in seconds) and the number of spikes recorded. Including this information in Table 1 for each afferent would be helpful.

Line 143: The sample sizes are quite limited ($N = 5-6$ for each of the three main LTMR types). The conclusion that club endings differ from Merkel and lanceolate endings in their touch versus whisking responsivity hinges on a small number of cells with W/WT properties. This makes the reliability of the classification (as noted above) particularly critical.

The whisking responses in this study are elicited via electrical stimulation of the whisking muscles. While this method effectively mimics the bending forces associated with whisker-object contact, it only partially replicates the forces present during natural whisking. Thus, while the evidence for touch responses in club-like endings is convincing, the conclusion that these endings do not respond to self-motion during natural whisking is less so. Prior work (e.g., Severson et al., 2017) has shown that most afferents respond to self-motion during natural whisking. If club-like endings also respond to self-motion, this would challenge the rationale for seeking an anatomical basis for selective touch responses in the latter half of the manuscript. While intra-axonal recordings in awake, head-fixed animals may not be feasible, a paragraph acknowledging this limitation should be added to the Discussion.

Minor Comments

The Introduction focuses exclusively on artificial whisking. Studies examining mechanoreceptor function during natural whisking should be explicitly discussed in both the Introduction and Discussion.

Line 457: If a W cell is defined as having $\text{abs}(TI) < 0.2$, should the range for a WT cell not be -0.2 to 0.8 ?

Figure 3: The PSTHs are missing time scales.

Line 137: The latency difference (~ 100 ms) between the two example T cells is surprisingly large. Could the authors clarify how this is possible?

Reviewer #3

(Remarks to the Author)

This is an exciting study which brings to bear the most elegant methodologies of morphological anatomy and electrophysiology into a sensory paradigm that is, as much as can be achieved given the goals, a naturalistic and behaviorally relevant one – whisking and touching. The authors have identified the termination structure of tactile afferents from the rat whisker follicle, confirming the functional properties of previously identified classes and, beyond that, working out a class of receptor until now seen anatomically but never functionally documented. This latter class are Club-like Receptors and, as stated straight off in the title, these respond to light touch but not to whisking. Though the number that could be studied was limited to 5 due to the delicateness of the experimental setup, those sampled receptors appear to stand out qualitatively as having a distinct functional repertoire. Finally, the authors examine the physical substrate in which the receptors are embedded and provide a convincing account for how this substrate might isolate them from the mechanical inputs of whisking while leaving them exquisitely sensitive to the mechanical inputs that occur when the moving whisker collides with an object.

The text is clearly written and illustrated.

Overall, this work will be a significant contribution to sensory neuroscience.

While this reviewer has no criticisms of methodologies or results, I believe that the findings would be strengthened by including somewhat broader perspectives on some points.

First, with regard to the behavioral role of whisking, the authors have broken down behavior into active versus passive: “Natural perception is primarily active – animals move and move their sensory organs as part of perceiving their environments.” This is somewhat limiting and perhaps not entirely accurate. Just as humans carry out many tactile judgments by holding the fingertips immobile – thus for instance the potter lightly touches their hands against the rotating clay to assess its consistency, moisture, and symmetry, using tactile feedback from the moving surface. Rubbing ones hands against the clay is not the optimal sensorimotor strategy under some conditions. Likewise, among freely moving rodents the whisking mode of perception is observed more frequently, but this is because experimentalists less often create conditions in which tactile information can be acquired by the immobile whiskers. Recent work has emphasized the dual modes of

vibrissal information acquisition: generative, where the animal sweeps its whiskers forward and backwards to palpate objects, and receptive, where the animal keeps its whiskers still, and collects mechanical signals from the motion of an object such as a vibrating surface (Diamond & Arabzadeh, 2013; Diamond & Toso, 2023; Fassihi et al., 2014). The dichotomy generative vs receptive is far preferable to active vs passive because (1) holding receptor surfaces stable (finger tips or whiskers) is hardly a passive task from the neuromuscular point of view, but is actually quite complex, and (2) the sensory nervous system is not “passive” during receptive sensing, but rather is highly engaged in optimizing information acquisition and likely is actively forming predictive models. In my view, the present study would be better presented in the generative/receptive framework, not the active/passive framework, also discussing the relevant work.

Second, as the authors well know, the whisking cycle in freely moving rats relies on the coordinated action of intrinsic and extrinsic muscles within the mystacial pad (Berg & Kleinfeld, 2003; Hill et al., 2011). Intrinsic muscles, located entirely within the pad, insert on individual vibrissae and are primarily responsible for generating protraction movements of the whiskers. Extrinsic muscles, such as the nasolabialis and maxillolabialis, originate outside the pad and provide broader support, contributing to retraction of whisker positioning. The interplay between these two muscle groups ensures precise control of vibrissa motion, enabling rats to generatively sample their tactile environment during exploration. For reasons that are not entirely clear (at least to me), electrical whisking produces only protraction (Arabzadeh et al., 2005). As noted in the present manuscript, “retraction was passive.” The implication is that, because the electrical whisking protocol did not produce a full, natural protraction AND RETRACTION, authors should be more circumspect in designating the Club-like Receptors as being receptive only to touch. It is possible, even if not likely, that the club receptors could fire during whisker retraction if that retraction were active rather than passive. The receptors might be sensitive to extrinsic muscle contraction.

Mathew Diamond

Arabzadeh, E., Zorzin, E., & Diamond, M. E. (2005). Neuronal encoding of texture in the whisker sensory pathway. *PLoS Biol*, 3(1), e17. <https://doi.org/10.1371/journal.pbio.0030017>

Berg, R. W., & Kleinfeld, D. (2003). Rhythmic Whisking by Rat: Retraction as Well as Protraction of the Vibrissae Is Under Active Muscular Control. *Journal of Neurophysiology*, 89(1), 104-117. <https://doi.org/10.1152/jn.00600.2002>

Diamond, M. E., & Arabzadeh, E. (2013). Whisker sensory system—From receptor to decision. *Progress in Neurobiology*, 103, 28-40. <https://doi.org/doi:10.1016/j.pneurobio.2012.05.013>

Diamond, M. E., & Toso, A. (2023). Tactile cognition in rodents. *Neuroscience & Biobehavioral Reviews*, 105161. <https://doi.org/https://doi.org/10.1016/j.neubiorev.2023.105161>

Fassihi, A., Akrami, A., Esmaeili, V., & Diamond, M. E. (2014). Tactile perception and working memory in rats and humans. *Proceedings of the National Academy of Sciences*, 111(6), 2331-2336. <https://doi.org/10.1073/pnas.1315171111>

Hill, D. N., Curtis, J. C., Moore, J. D., & Kleinfeld, D. (2011). Primary motor cortex reports efferent control of vibrissa motion on multiple timescales. *Neuron*, 72(2), 344-356. [https://doi.org/S0896-6273\(11\)00871-3](https://doi.org/S0896-6273(11)00871-3) [pii] 10.1016/j.neuron.2011.09.020

Reviewer #4

(Remarks to the Author)

Muramoto and colleagues describe a specific class of mechanoreceptors that responds to touch, but not whisking in air, by recording from their axons, characterizing their responses, and reconstructing their morphology.

The study is important for the field of vibrissa biophysics and has implications for understanding the low-level coding of whisker touch coding in rodents. The findings seem to be well supported by the data, and the presentation is clear.

My only requests concern slight clarifications and discussion points and do not warrant another round of review.

The argument that the club-like receptors avoid activation during whisking in air due to their location is convincing, but this logic does not immediately explain how they still respond to whisker touch, which should, at least in one major component, contain torque along the whisker shaft that acts on a similar point of rotation in the follicle as the momentum and air resistance during whisking in air. Are the club-like receptor expected to predominantly activate to axial whisker deflections, lateral non-bending loads, or high-frequency vibrations or stick-clip events maybe? It would be interesting to at least add a sentence or two discussing this.

Related to this, a bit more information about the nature of the object contacts in the main text would be useful to readers to make it easier to interpret the findings. For example, what kind of whisker deflections were observed, at what speeds, etc.

All in all, this is a solid contribution to the literature and is improving our understanding of the vibrissa system.

Version 1:

Reviewer comments:

Reviewer #1

(Remarks to the Author)

The authors addressed my concerns. This is an outstanding paper. I support publication.

Reviewer #2

(Remarks to the Author)

Concerns about sample size and reliability addressed through an elegant and convincing analysis.

Reviewer #3

(Remarks to the Author)

In my first-round review I pointed out the merits of this study so I omit that form of summary in the second-round review. All of the queries raised by me and by the other reviewers (who I congratulate for the preciseness of their observations) have been addressed to satisfaction.

Reviewer #4

(Remarks to the Author)

The revised manuscript addresses my comments about clarifying the degree to which the results can be used to infer a mechanism for how touch but not whisking selective responses could arise and how far the findings can be expected to generalize. The revised version also addresses related comments made by other reviewers. In my opinion this can be published as is.

Club-like Receptors Respond to Light Touch but no to Whisking

Point by point reply

Dear Reviewers,

We are deeply grateful for the fair and constructive comments and suggestions that have helped us improve our manuscript.

Below are the Reviewer's comments, in black, and our replies, in blue, starting with ">>>Reply:". Please note that all line numbers refer to the revised manuscript when viewed in "All Markup" mode (PDF).

REVIEWER COMMENTS

Reviewer #1 (Remarks to the Author):

In their paper 'Club-like Receptors Respond to Light Touch but not to Whisking' Muranoto et al. investigate structure function relationships of vibrissa afferents. Specifically, the authors apply intra-axonal recordings combined with neurobiotin-labeling, electron-microscopy and electric whisking paradigms, an astounding range of analysis technologies.

The authors identify a range of afferents including significant numbers of vibrissal Merkel-receptors, lanceolate-endings and club-like endings. The highlight of the study is the author's conclusion that club-like endings are so-called touch cells, i.e. cells that do not respond to whisking alone but are responsive to object contact.

The authors also do beautiful electron-microscopy work and reveal novel details about the anchoring of club-like endings, which they show to be attached to the glassy membrane and also being anchored by collagen fibers. On grounds of these morphological reconstructions and by based on additional comparative evidence (comparisons to non-whisking cats) the authors argue that the ringwulst might actually contribute to dampening whisking related responses in club-like afferents. I think the whisking-response-dampening by the ringwulst is an interesting hypothesis, but the authors present it as a fact, which I think is putting this idea too strongly.

>>>Reply

We thank the Reviewer for their positive feedback on our work. In response to their comment, we have revised the text to present the dampening of whisking responses by the ringwulst as a hypothesis rather than a confirmed fact. The specific changes are detailed in our reply to the Reviewer's comment below.

Major Points

1. This is an important paper. The authors are pioneers in understanding the vibrissa follicle and whisker response properties. The intra-axonal recordings performed here in whisking animals are nontrivial, but absolutely decisive for understanding the vibrissal system. Classifying club-like endings as touch-cells is an important advance that fully justifies publication.

>>>Reply

We thank the Reviewer for appreciating our efforts and recognizing the value of our results.

2. What do the authors think is origin of the prominent subthreshold activity seen in many afferents (Figure 2)? I am asking, because the authors penetrate cells a centimeter or more away from the mechanosensory tip of the afferents and I expected that we should see little subthreshold signal as a result of axonal filtering.

>>>Reply

We agree – only little subthreshold signal was expected to be seen in these recordings. We thus believe that the subthreshold fluctuations seen in our recordings result from noise.

We distinguish between two noise components: low-frequency (< 20 Hz) and high-frequency. Based on our accumulated experience, the primary source of the low-frequency component is physiological, mostly cardiac and respiratory rhythms. The primary source of the high-frequency component is likely electrical noise – both inductive (RF) and additive (inherent to the electronic devices).

We now add a comment clarifying this in Lines 315-317: “Recorded signals included low-frequency components (<20 Hz), likely reflecting physiological activity such as cardiac and respiratory rhythms, and high-frequency components, primarily attributed to neural electrical activity. Recorded data were analyzed....”

3. I think the ms will benefit if the results followed a bit more tightly the Figures and individual panels. In general, the paper is well written, but the detailed link to the Figures would often be advantageous.

>>>Reply

We thank the Reviewer for this note. We now modified the text such that it follows more tightly the Figures and individual panels. Changes were introduced in the following lines: 99, 101, 105, 108, 111-113, 120-121, 141, 144 and 147-148

4. As I said earlier the whisking-response-dampening by the ringwulst is an interesting hypothesis. It would be great if the authors could explain thinking in more detail, perhaps adding a schematic as to how the ringwulst is supposed to do this trick. Also, I am not sure the idea is correct and the authors need to present this more cautiously.

>>>Reply

We thank the reviewer for highlighting this important point. We agree that the idea of inertial damping by the ringwulst (Rw) is indeed a hypothesis and should be presented as such. In the revised manuscript, we have taken two steps to address the reviewer's suggestion:

1. Clarified that this is a hypothesis: We have revised the relevant sections (Results and Discussion) to explicitly state that the whisking-response-dampening function of the Rw is a proposed mechanism based on morphological observations. The wording has been made more cautious to reflect the speculative nature of this interpretation. Our changes are (the changes addressing the Reviewer's comment are underlined):
 - a. Line 152-158, changed text to read: The third factor found to potentially contribute to the decoupling of club-like endings from tissue deformations during whisking in air was the structure of the Rw in the rat and its attachment to the glassy membrane surrounding the epithelial follicle. The Rw is suspended from a curved edge of the glassy membrane (Fig. 6). The mass of the Rw determines the extent of inertial damping of the movements during whisking in air: the larger the mass the smaller the movement at the neck of the Rw, where the club-like endings are located, should be.
 - b. Lines 197-199, changed text to read: We further hypothesize that adding extra care of Schwann protection and inertial damping by the Rw likely complete the supreme isolation of club-like ending from whisking in air deformations.
2. We better explain the hypothesis and provide a suitable reference from engineering:
 - a. Lines 158-164: we have added the following text: "This damping effect is analogous to that achieved using pendulum-type mass dampers in engineering applications, such as those used to suppress vibrations in tall buildings²⁵. The operating principle involves suspending a mass (forming a pendulum) within the structure, allowing it to swing freely. When the structure undergoes lateral motion, the pendulum oscillates out of phase with the movement, thereby exerting an inertial counterforce that reduces the amplitude of the structural vibrations (see Supplementary Movie 2)."
 - b. We added an illustrative movie as Supplementary Movie 2 as below.
Supplementary Movie 2. Whisking-response-damping hypothesis.
URL: <https://www.dropbox.com/scl/fi/t3cwpv4vaj6o3549l9ri2/Supplementary-Movie-02-Taiga-Muramoto.mp4?rlkey=2ix1gzoyjvsfr7c71j3uo3ohu&st=f6qcdo09&dl=0>

This video shows a simple thick-paper-based mechanical model demonstrating how vibration is differentially transmitted depending on structural support. Two elevated platforms receive floor vibrations, but only one supports an inertial mass. The green plastic cap on the damped platform moves less, indicating effective vibration damping²⁵ (Young et al., 2022). This setup is used to illustrate our hypothesis: the inertial mass represents the ringwulst (black circle with red bar), the yellow triangle corresponds to a club-like mechanoreceptor endings, the sky-blue surface represents the glassy membrane, and the dark-blue base symbolizes the follicle's capsule. Movement of the capsule—analogue to whisker follicle motion during active whisking—is transmitted via a green dashed strip, modeling connective tissue. The reduced movement of the green plastic cap exemplifies how the ringwulst might dampen whisking-related vibrations, allowing club-like endings to selectively respond to touch.

We believe these changes improve the clarity and transparency of the hypothesis, and we hope they address the reviewer's concern.

5. I enjoyed the careful electron-microscopy.

>>>Reply

Thank you.

Minor

1. Make sure all abbreviations are explained in all legends; I could not find IMZ.

>>>Reply

Thank you for the careful check. We went through all abbreviations and verified that they are explained. As for IMZ, please look at the legend of Figure 4: "IMZ: intermediary zone".

2. Figure 5 C, D please show higher magnification of the immunofluorescence.

>>>Reply

We have added two panels: h and i. Panel h shows a higher magnification of panel d, while panel i demonstrates the striated regularity of the collagen fibers in the thick bundles of the Rw, as revealed by electron microscopy.

We added text in Lines 147-148: The bundles were collagen I-immunopositive and showed a typical banding pattern (Fig. 5d, h, i). We modified the figure legend accordingly.

Reviewer #2 (Remarks to the Author):

This study is commendable for tackling a fundamental and unresolved question in sensory neuroscience: the structure-function relationships in low-threshold mechanoreceptors (LTMRs). The authors employ a technically demanding approach – intra-axonal recordings – to address this issue, and they present valuable data on whisker LTMR responses during artificial whisking, along with anatomical insights.

>>>Reply

We thank the Reviewer for appreciating our efforts and recognizing the value of our results.

However, I have concerns regarding the sample size and the generalizability of the findings to natural whisking behavior.

>>>Reply

We appreciate the reviewer for highlighting this important point. We have addressed all of your comments—please refer to the point-by-point responses below.

Major Comments

Line 94: Were the recordings of sufficient duration to yield robust estimates of the response index for each afferent? Depending on the number of spikes in each recording, the classification of an afferent could potentially shift based on the presence or absence of few spikes. Please provide more specific information on the duration of each

recording (in seconds) and the number of spikes recorded. Including this information in Table 1 for each afferent would be helpful.

>>>Reply

We now include this information in Table 1. We have added two columns, one for recording time and the second for the total number of recorded spikes. In addition, in order to address the justified concern of the Reviewer that the classification might not be reliable enough due to the small number of spikes, we conducted a permutation analysis and added its results in a 3rd new column – “classification reliability”. In this analysis we repeated the classification 10 times, each time with only one train in air and one train against an object, randomly selected. All neurons except one (Cell #18) showed high reliability (showing the same classification in at least 80 % of the permutations). Cell #18 showed equal probability to be classified as W or WT, hence its class was modified to “W / WT” in the table.

We have added the following text in Lines 344-347. “Classification reliability was evaluated by repeating the classification process 10 times, each time with only one train in air and one train against an object, randomly selected. Classification reliability (Table 1) was defined as the fraction of identical classifications among the 10 repetitions.” We also modified Table 1 and the text in Lines 620-621.

We thank the Reviewer for the comment leading to this important analysis.

Line 143: The sample sizes are quite limited (N = 5-6 for each of the three main LTMR types). The conclusion that club endings differ from Merkel and lanceolate endings in their touch versus whisking responsivity hinges on a small number of cells with W/WT properties. This makes the reliability of the classification (as noted above) particularly critical.

>>>Reply

We acknowledge the validity of this concern and have taken steps to address it. We have thus performed the reduced dataset analysis described in our Reply to the previous point. This analysis revealed that except for one cell (#18) our classifications are reliable, meaning that they are not expected to change during prolonged recordings.

We now describe this in the text (Lines 344-347 [[same lines as in previous comment]]). We have also changed the class of cell #18 from “W” to “W / WT”.

The whisking responses in this study are elicited via electrical stimulation of the whisking muscles. While this method effectively mimics the bending forces associated with whisker-object contact, it only partially replicates the forces present during natural whisking. Thus, while the evidence for touch responses in club-like endings is convincing, the conclusion that these endings do not respond to self-motion during natural whisking is less so. Prior work (e.g., Severson et al., 2017) has shown that most afferents respond

to self-motion during natural whisking. If club-like endings also respond to self-motion, this would challenge the rationale for seeking an anatomical basis for selective touch responses in the latter half of the manuscript. While intra-axonal recordings in awake, head-fixed animals may not be feasible, a paragraph acknowledging this limitation should be added to the Discussion.

>>>Reply

We acknowledge the relevance and validity of this point.

First, we have modified the major selectivity statement in the Discussion:

Lines 186-189: the statement now reads: “Here we revealed their unique role in the encoding of active touch – during whisker protraction, club-like receptors respond selectively to active contacts and ignore whisking active protraction in air. We revealed it using artificial whisking, in which protraction is active and retraction is passive²⁹ (Arabzadeh et al 2005).”

In addition, in response to the first Minor Comment below, we have added several paragraphs to both the Introduction and Discussion. These paragraphs describe mechanoreceptor function during self-generated whisking in awake animals and further emphasize the differences between natural and artificial whisking, which underlie the Reviewer’s justified concern. Please see the revised sections below.

Minor Comments

The Introduction focuses exclusively on artificial whisking. Studies examining mechanoreceptor function during natural whisking should be explicitly discussed in both the Introduction and Discussion.

>>>Reply

Thank you for this note. We modified both the Introduction and Discussion accordingly.

Introduction: added

(Lines 71-74) “During generative sensing, the mechanical activation of the whisker follicle is shaped by a combination of kinematic and force-related variables—such as whisker angle, velocity, and bending moment—making the selective coding of specific stimulus features challenging^{9,10,11} (Campagner et al 2016, Campagner et al 2018, Bush et al 2016).”

(Lines 77-80): “Recordings from identified Merkel afferents during self-generated whisking in awake mice showed that Merkel receptors encode both whisking and touch, with some afferents responding predominantly to touch, some to whisking motion, and others to both¹⁷ (Severson et al, 2017)”.

Discussion: added a new paragraph (Lines 212-222): “The whisker azimuth is also

encoded via a continuous representation of the whisking phase by primary afferents^{7,33} (Szwed et al 2003; Wallach et al 2016). In awake mice, whisking phase was shown to be encoded with high resolution by follicle-associated Merkel afferents, and with lower precision by cutaneous and supraorbital mechanoreceptors^{17,34} (Severson et al 2017, Severson et al 2019). Furthermore, in awake behaving rodents, the firing of primary afferents was found to be best predicted by a combination of kinematic (e.g., angle, velocity) and dynamic (e.g., bending moment) variables, making the direct interpretation of their responses ambiguous^{9,10} (Campagner et al 2016, Campagner et al 2018). Here, we propose that combining re-afferent (self-motion) signals from Merkel and other afferents with ex-afferent (contact-driven) signals from club-like receptors may help disambiguate azimuthal coding upon object contact. Whether rodents utilize either of these coding strategies, or a combination thereof, remains to be tested."

Line 457: If a W cell is defined as having $\text{abs}(\text{TI}) < 0.2$, should the range for a WT cell not be -0.2 to 0.8 ?

>>>Reply

Following Yu et al 2006 and Szwed et al 2003, we define WT cells as those with $0.2 \leq \text{TI} < 0.8$. The complementary range for negative TIs ($-0.8 \leq \text{TI} < -0.2$) was defined in Yu et al as W-T cells. Importantly, however, such W-T cells were not observed so far among primary afferents, hence they are not discussed here.

Figure 3: The PSTHs are missing time scales.

>>>Reply

The scale bar for all subplots is depicted at the top right corner of the figure. This is now noted in the Figure Legend (Line 561).

Line 137: The latency difference (~ 100 ms) between the two example T cells is surprisingly large. Could the authors clarify how this is possible?

>>>Reply

We thank the Reviewer for this comment. Indeed cell No. 3 responds to the onset of contact ("Contact cell" in Szwed et al 2003) whereas cell No. 8 responds to detachment ("Detach cell" in Szwed et al 2003). We now add this information in the paper:

Lines 112-115: "Note that (Fig. 2) cell No.3 responded to the onset of contact, "Contact T cell"⁷ (Szwed et al 2003) whereas cell No.8 responded to detachment, "Detach T cell"⁷. Herein we do not distinguish between different subtypes of T cells."

Reviewer #3 (Remarks to the Author):

This is an exciting study which brings to bear the most elegant methodologies of morphological anatomy and electrophysiology into a sensory paradigm that is, as much as can be achieved given the goals, a naturalistic and behaviorally relevant one – whisking and touching. The authors have identified the termination structure of tactile afferents from the rat whisker follicle, confirming the functional properties of previously identified classes and, beyond that, working out a class of receptor until now seen anatomically but never functionally documented. This latter class are Club-like Receptors and, as stated straight off in the title, these respond to light touch but not to whisking. Though the number that could be studied was limited to 5 due to the delicateness of the experimental setup, those sampled receptors appear to stand out qualitatively as having a distinct functional repertoire. Finally, the authors examine the physical substrate in which the receptors are embedded and provide a convincing account for how this substrate might isolate them from the mechanical inputs of whisking while leaving them exquisitely sensitive to the mechanical inputs that occur when the moving whisker collides with an object.

The text is clearly written and illustrated.

Overall, this work will be a significant contribution to sensory neuroscience.

While this reviewer has no criticisms of methodologies or results, I believe that the findings would be strengthened by including somewhat broader perspectives on some points.

>>>Reply

Dear Dr. Mathew Diamond,

We greatly appreciate your valuable and thoughtful review. We have addressed all of your comments—please refer to the point-by-point responses below.

First, with regard to the behavioral role of whisking, the authors have broken down behavior into active versus passive: “Natural perception is primarily active – animals move and move their sensory organs as part of perceiving their environments.” This is somewhat limiting and perhaps not entirely accurate. Just as humans carry out many tactile judgments by holding the fingertips immobile – thus for instance the potter

lightly touches their hands against the rotating clay to assess its consistency, moisture, and symmetry, using tactile feedback from the moving surface. Rubbing ones hands against the clay is not the optimal sensorimotor strategy under some conditions. Likewise, among freely moving rodents the whisking mode of perception is observed more frequently, but this is because experimentalists less often create conditions in which tactile information can be acquired by the immobile whiskers. Recent work has emphasized the dual modes of vibrissal information acquisition: generative, where the animal sweeps its whiskers forward and backwards to palpate objects, and receptive, where the animal keeps its whiskers still, and collects mechanical signals from the motion of an object such as a vibrating surface (Diamond & Arabzadeh, 2013; Diamond & Toso, 2023; Fassihi et al., 2014). The dichotomy generative vs receptive is far preferable to active vs passive because (1) holding receptor surfaces stable (finger tips or whiskers) is hardly a passive task from the neuromuscular point of view, but is actually quite complex, and (2) the sensory nervous system is not “passive” during receptive sensing, but rather is highly engaged in optimizing information acquisition and likely is actively forming predictive models. In my view, the present study would be better presented in the generative/receptive framework, not the active/passive framework, also discussing the relevant work.

>>>Reply

We thank Dr. Diamond for this very clear presentation of the motor-sensory modes of behavior. We agree with the Reviewer that the generative/receptive division is an excellent view over vibrissal information acquisition. We have thus modified the text accordingly:

Line 50: Replaced “move” with “control”

Lines 55-58: added “Recent work has emphasized the dual modes of vibrissal information acquisition: generative, where the animal sweeps its whiskers forward and backwards to palpate objects, and receptive, where the animal keeps its whiskers still, and collects mechanical signals from the relative motion between the head and an object⁵ (Diamond & Toso, 2023).”

Line 59: Replaced “Active” with “Generative”.

Second, as the authors well know, the whisking cycle in freely moving rats relies on the coordinated action of intrinsic and extrinsic muscles within the mystacial pad (Berg & Kleinfeld, 2003; Hill et al., 2011). Intrinsic muscles, located entirely within the pad, insert on individual vibrissae and are primarily responsible for generating protraction movements of the whiskers. Extrinsic muscles, such as the nasolabialis and

maxillolabialis, originate outside the pad and provide broader support, contributing to retraction of whisker positioning. The interplay between these two muscle groups ensures precise control of vibrissa motion, enabling rats to generatively sample their tactile environment during exploration. For reasons that are not entirely clear (at least to me), electrical whisking produces only protraction (Arabzadeh et al., 2005). As noted in the present manuscript, “retraction was passive.” The implication is that, because the electrical whisking protocol did not produce a full, natural protraction AND RETRACTION, authors should be more circumspect in designating the Club-like Receptors as being receptive only to touch. It is possible, even if not likely, that the club receptors could fire during whisker retraction if that retraction were active rather than passive. The receptors might be sensitive to extrinsic muscle contraction.

>>>Reply

Once more, we thank Dr. Diamond for this important observation. Indeed, our results refer primarily to active protraction. We have thus modified the selectivity statement in the Discussion to reflect this reservation:

Lines 186-189: the statement now reads: “Here we revealed their unique role in the encoding of active touch – during whisker protraction, club-like receptors respond selectively to active contacts and ignore whisking active protraction in air. We revealed it using artificial whisking, in which protraction is active and retraction is passive²⁹(Arabzadeh et al 2005).”

Mathew Diamond

Arabzadeh, E., Zorzin, E., & Diamond, M. E. (2005). Neuronal encoding of texture in the whisker sensory pathway. *PLoS Biol*, 3(1), e17.

<https://doi.org/10.1371/journal.pbio.0030017>

Berg, R. W., & Kleinfeld, D. (2003). Rhythmic Whisking by Rat: Retraction as Well as Protraction of the Vibrissae Is Under Active Muscular Control. *Journal of Neurophysiology*, 89(1), 104-117. <https://doi.org/10.1152/jn.00600.2002>

Diamond, M. E., & Arabzadeh, E. (2013). Whisker sensory system - From receptor to decision. *Progress in Neurobiology*, 103, 28-40.

<https://doi.org/doi:10.1016/j.pneurobio.2012.05.013>

Diamond, M. E., & Toso, A. (2023). Tactile cognition in rodents. *Neuroscience & Biobehavioral Reviews*, 105161.

<https://doi.org/https://doi.org/10.1016/j.neubiorev.2023.105161>

Fassihi, A., Akrami, A., Esmaeili, V., & Diamond, M. E. (2014). Tactile perception and working memory in rats and humans. *Proceedings of the National Academy of Sciences*, 111(6), 2331-2336. <https://doi.org/10.1073/pnas.1315171111>

Hill, D. N., Curtis, J. C., Moore, J. D., & Kleinfeld, D. (2011). Primary motor cortex reports efferent control of vibrissa motion on multiple timescales. *Neuron*, 72(2), 344-356.

[https://doi.org/S0896-6273\(11\)00871-3](https://doi.org/S0896-6273(11)00871-3) [pii]
10.1016/j.neuron.2011.09.020

Reviewer #4 (Remarks to the Author):

Muramoto and colleagues describe a specific class of mechanoreceptors that responds to touch, but not whisking in air, by recording from their axons, characterizing their responses, and reconstructing their morphology.

The study is important for the field of vibrissa biophysics and has implications for understanding the low-level coding of whisker touch coding in rodents. The findings seem to be well supported by the data, and the presentation is clear.

My only requests concern slight clarifications and discussion points and do not warrant another round of review.

>>>Reply

We greatly appreciate this supportive and constructive review. Please see below our point-by-point replies.

The argument that the club-like receptors avoid activation during whisking in air due to their location is convincing, but this logic does not immediately explain how they still respond to whisker touch, which should, at least in one major component, contain torque along the whisker shaft that acts on a similar point of rotation in the follicle as the momentum and air resistance during whisking in air. Are the club-like receptor expected to predominantly activate to axial whisker deflections, lateral non-bending loads, or high-frequency vibrations or stick-slip events maybe? It would be interesting to at least add a sentence or two discussing this.

>>>Reply

We thank the Reviewer for this important point. We have added a paragraph in the Discussion that elaborates on the possible manners in which object-contact would differ than air resistance in shaft-follicle mechanics, as suggested by the Reviewer

The new paragraph (Lines 223-229): “The precise arrangement of club-like endings may be instrumental in understanding their mechanical activation during contact. The torque developed along the whisker shaft during contact or during whisking against air resistance is largely similar³⁵ (Hartmann, M. J. Z. 2011). Is the selectivity of club-like endings to contact determined solely by their exact location? Or is it also influenced by a sensitivity to axial whisker deflections, lateral non-bending loads, high-frequency vibrations, or stick-slip events³⁵? These questions remain to be clarified in future experiments, aided by precise tracking of whisker kinematics.”

Related to this, a bit more information about the nature of the object contacts in the main text would be useful to readers to make it easier to interpret the findings. For example, what kind of whisker deflections were observed, at what speeds, etc.

>>>Reply

We agree that this information would indeed be valuable. However, due to the considerable technical challenges associated with intra-axonal recordings during active whisking, we did not measure whisker kinematics in the present study.

We concur that future experiments should aim to track and quantify whisker kinematics as precisely as possible—primarily to help differentiate between the alternative interpretations raised by the Reviewer in the preceding comment.

We have added this point in the same paragraph of the Discussion:

Lines 228-229: “These questions remain to be clarified in future experiments, aided by precise tracking of whisker kinematics.”

All in all, this is a solid contribution to the literature and is improving our understanding of the vibrissa system.

>>>Reply

Once more we thank the Reviewer for this supportive and constructive review.

Other Changes

1) AUTHORS: Takahiro Furuta should become one of corresponding authors (✉) as mentioned above in the cover letter. And an additional coauthor: Masaaki Kitada³, who contributed serial semi-thin EM reconstruction and analysis, was newly added as shown below. Consequently, his name was removed from the acknowledgments. The all coauthors accepted the changes (c.f. consent forms).

Taiga Muramoto¹

Takahiro Furuta^{2✉}

Taro Koike³

Knarik Bagdasarian⁴

Sotatsu Tonomura⁵

Aya Takenaka²

Yosky Kataoka⁶

Mitsuyo Maeda⁷

Asami Eguchi⁶

Masaaki Kitada³

Kenzo Kumamoto¹

Ehud Ahissar⁴✉

Satomi Ebara¹✉

2) Line 20: “Osaka University” should be corrected as “The University of Osaka”.

3) Line 23: “Kawasaki Medical University” should be “Kawasaki Medical School”.

4) Lines 140-142: We added “embedded within a dense network of fine collagen fibers occupying core region of the C-shaped Rw (Figs. 5a-e, f-g, i, 6, Supplementary Movie 1).” to clarify the detailed morphology of the ending.

5) REFERENCES: Lines 403-501: We added 9 references: No.5, 9, 10, 11, 25, 29, 33, 34 and 35. And we modified all reference numbers accordingly.

6) METHODS: Lines 371-373: We used goat anti-collagen type I IgG for primary antibody and anti-goat IgG for second antibody. All the affected sections have been corrected: “The follicles were processed immunohistochemically using goat collagen type I IgG antibodies (1:200, Vector, CA, USA), Alexa Fluor 488 conjugated anti-goat IgG (1:300, Vector, USA) and pyridinium iodide for nuclei.”

7) ACKNOWLEDGEMENTS: Line 508: Corrected to “The University of Osaka”. And the following two acknowledgements are added: Line 512-513: Mr. Katsuhiko Taki (Nihon Visual Science Inc., Japan) for imaging technical 512 support, and 518: and Kobayashi Foundation (384) to S.E.. Because their contributions were significant.

8) Some typos were fixed: Line 133: angle, Line 512: University, Fig. 1: Pipette, and A/D converter.

Club-like Receptors Respond to Light Touch but no to Whisking

Point by point reply

ROUND 2

REVIEWERS' COMMENTS

Reviewer #1 (Remarks to the Author):

The authors addressed my concerns. This is an outstanding paper. I support publication.

>>>Reply

Thank you very much. We greatly appreciate this constructive review.

Reviewer #2 (Remarks to the Author):

Concerns about sample size and reliability addressed through an elegant and convincing analysis.

>>>Reply

Thank you very much. We greatly appreciate this constructive review.

Reviewer #3 (Remarks to the Author):

In my first-round review I pointed out the merits for this study so I omit that form of summary in the second-round review.

All of the queries raised by me and by the other reviewers (who I congratulate for the preciseness of their observations) have been addressed to satisfaction.

>>>Reply

Thank you very much. We greatly appreciate this constructive review.

Reviewer #4 (Remarks to the Author):

The revised manuscript addresses my comments about clarifying the degree to which the results can be used to infer a mechanism for how touch but not whisking selective responses could arise and how far the findings can be expected to generalize. The revised version also addresses related comments made by other reviewers. In my opinion this can be published as is.

>>>Reply

Thank you very much. We greatly appreciate this constructive review.